# Mechanical Characterisation and Numerical Modelling of TPMS-Based Gyroid and Diamond Ti6Al4V Scaffolds for Bone Implants: An Integrated Approach for Translational Consideration

**DOI:** 10.3390/bioengineering9100504

**Published:** 2022-09-24

**Authors:** Seyed Ataollah Naghavi, Maryam Tamaddon, Arsalan Marghoub, Katherine Wang, Behzad Bahrami Babamiri, Kavan Hazeli, Wei Xu, Xin Lu, Changning Sun, Liqing Wang, Mehran Moazen, Ling Wang, Dichen Li, Chaozong Liu

**Affiliations:** 1Institute of Orthopaedic & Musculoskeletal, Division of Surgery & Interventional Science, University College London, Royal National Orthopaedic Hospital, Stanmore, London HA7 4LP, UK; 2Department of Mechanical Engineering, University College London, London WC1E 7JE, UK; 3Aerospace and Mechanical Engineering Department, The University of Arizona, Tucson, AZ 85721, USA; 4National Engineering Research Center for Advanced Rolling and Intelligent Manufacturing, Institute of Engineering Technology, University of Science and Technology Beijing, Beijing 100083, China; 5State Key Laboratory for Manufacturing Systems Engineering, School of Mechanical Engineering, Xi’an Jiaotong University, Xi’an 710049, China; 6National Medical Products Administration (NMPA) Key Laboratory for Research and Evaluation of Additive Manufacturing Medical Devices, Xi’an Jiaotong University, Xi’an 710054, China; 7State Key Laboratory of Metal Matrix Composites, School of Materials Science and Engineering, Shanghai Jiao Tong University, Shanghai 200240, China

**Keywords:** additive manufacturing, mechanical properties, bending strength, torsional strength, lattice structures, biomedical scaffolds, bone scaffolds, Ti6Al4V scaffolds, TPMS scaffolds, finite element analysis

## Abstract

Additive manufacturing has been used to develop a variety of scaffold designs for clinical and industrial applications. Mechanical properties (i.e., compression, tension, bending, and torsion response) of these scaffolds are significantly important for load-bearing orthopaedic implants. In this study, we designed and additively manufactured porous metallic biomaterials based on two different types of triply periodic minimal surface structures (i.e., gyroid and diamond) that mimic the mechanical properties of bone, such as porosity, stiffness, and strength. Physical and mechanical properties, including compressive, tensile, bending, and torsional stiffness and strength of the developed scaffolds, were then characterised experimentally and numerically using finite element method. Sheet thickness was constant at 300 μm, and the unit cell size was varied to generate different pore sizes and porosities. Gyroid scaffolds had a pore size in the range of 600–1200 μm and a porosity in the range of 54–72%, respectively. Corresponding values for the diamond were 900–1500 μm and 56–70%. Both structure types were validated experimentally, and a wide range of mechanical properties (including stiffness and yield strength) were predicted using the finite element method. The stiffness and strength of both structures are comparable to that of cortical bone, hence reducing the risks of scaffold failure. The results demonstrate that the developed scaffolds mimic the physical and mechanical properties of cortical bone and can be suitable for bone replacement and orthopaedic implants. However, an optimal design should be chosen based on specific performance requirements.

## 1. Introduction

Additive manufacturing (AM) or 3D printing is a technique that builds 3D objects from a 3D digital model in a layer-by-layer fashion. This is accomplished either using computer-aided design (CAD) or scanning the object. Over the past decades, AM technology has been used in many fields, such as the medical, automotive, aerospace, and marine industries [1,2,3,4,5,6] and for personal protective devices [7]. Advances in AM have opened new possibilities for the fabrication of biomedical devices or constructs with synergistic biological and mechanical properties that can mimic natural tissue structures and the physiological environment. Additionally, AM has been utilized in the medical field to produce vascularized tissues and organs (via bioprinting) or to develop patient-specific orthopaedic and dental implants (via metal printing) [8]. The AM of metals (e.g., Ti-6% aluminium [AL]-4% vanadium [V] alloy, hereafter referred to as Ti6Al4V) has enabled the development of customized complex-shaped structures, including porous architecture with a high strength-to-weight ratio to be used as bone scaffolds for orthopaedic applications [9]. These scaffolds feature high surface areas that provide a template for initial cell attachment, proliferation, differentiation, and tissue formation. Their distinctive porous structure can enhance the osseointegration process and long-term biologic fixation [10,11].

Titanium alloy (Ti6Al4V) has been extensively used in orthopaedic implants due to its excellent biocompatibility and osseointegration. It also has high mechanical strength, stiffness, and good corrosive resistance properties [9,12]. Hip joint implants are made out of solid titanium alloy that has a much higher Young’s modulus of about 110 gigapascals (GPa) compared to bone (in the range of 6–30 GPa) [13,14]. This results in a stress shielding effect, provoking bone resorption and increasing the risk of bone fracture and the need for revision surgeries [15,16,17]. To prevent stress shielding, the Young’s modulus of the implants can be potentially reduced to match that of the bone. Based on the Gibson−Ashby model [18], mechanical properties of scaffolds are related to their porosity, where the Young’s modulus reduces as the porosity increases. Manufacturing such designs is now possible using AM techniques [19,20]. Large bone defects often result from musculoskeletal tumour resection, infection, or trauma that is unable to heal properly without surgical stabilisation [21].

Triply periodic minimal surfaces (TPMSs) are minimal surfaces which are periodic in three independent directions. They have emerged as an effective solution for the construction of porous structures in recent years and have gained attention in the field of tissue engineering scaffolds. The TPMS structures have a continuously curved surface, where they can avoid any localized stress concentration, and they also have a smooth stress distribution to their surrounding surfaces [22]. They have zero-mean curvature at every point on their surface, which enhances their load-bearing capacity and mechanical properties [23]. They also have improved biological features due to their large surface area, which can provide more effective fixation to the host tissue using enhanced initial cell seeding [24,25]. Every TPMS structure has a unique morphology where its morphological parameters, such as pore shape and size and strut thickness and porosity, can be controlled and adjusted to provide sufficient mechanical properties to support physiological loadings and match the required mechanical properties of local bone [26]. Increasing the porosity of porous scaffolds can improve cell growth and nutrient transport. However, this can also reduce the mechanical strength and stiffness of the scaffold. This is where a compromise is needed between the mechanical properties and the permeability of the scaffold during the design process [27,28]. Moreover, manufacturability and biocompatibility are other important characteristics of bone scaffolds to be considered [29]. Recent studies have investigated typical TPMS structures, including gyroid, diamond, primitive, and isometric wrapped surfaces. These studies have highlighted that gyroid structure has one of the highest levels of permeability that can enhance the settling speed of cells upon the static seeding of immortalized mesenchymal stem cells, while the diamond structure displays a high stiffness, surface area, and tortuosity [30,31]. Tortuosity concerns the flow path complexity and is defined as the ratio of the mean flow path length to the shortest distance between the flow path inlet and outlet planes.

Laser powder bed fusion is a popular AM technique for metals. Currently, different AM processes exist for biomedical applications, such as electron-beam melting and selective laser melting (SLM) [32,33]. A shortcoming of current AM technologies is that they cannot produce porous structures at the expected level of accuracy. Thus, originally designed features cannot be reproduced. The geometrical deviations can affect the osseointegration process and the mechanical properties of the scaffolds [34,35,36]. Such discrepancies can be considered in the design process to overcome the current manufacturing limitations.

To date, many studies have focused on obtaining mechanical properties and failure mechanisms of Ti6Al4V TPMS scaffolds using static compression, static tension, and fatigue tests [37,38,39,40,41,42,43]. However, it is known that scaffolds for bone replacement and orthopaedic implants also undergo bending and torsional loading in the body. Hence, it is crucial to investigate and quantify porous scaffolds’ bending and torsional properties for such applications. Currently, only a few studies have investigated the bending and torsional properties of metallic (titanium alloy, steel, and aluminum) and polyether−ether−ketone porous structures [44,45,46,47,48,49,50,51,52,53], while to the best of our knowledge, no study has investigated the performance of TPMS porous structures made of Ti6Al4V under bending and torsion. This study aimed to systematically characterise the interplay between different porosities of Ti6Al4V sheet TPMS gyroid and diamond scaffolds, and their respective compressive, tensile, flexural, and torsional properties. Data obtained in this paper are compared with human cortical bone mechanical properties and it is possible to modulate the design to adapt to particular applications such as large bone defects and load-bearing orthopaedic applications (i.e., porous hip implants that undergo multiple stress states).

## 2. Materials and Methods

### 2.1. Design and Manufacturing of Porous Titanium Scaffolds

#### 2.1.1. Micro-Structure-Driven Design of the Scaffolds

Two TPMS structures, that is, gyroid (G) and diamond (D) scaffolds, were developed using the nTopology software (version 3.25.3, New York, NY, USA—Figure 1a). The equations used to develop the TPMS Schoen gyroid (1) and Schwarz diamond lattice (2) unit cells are as follows: [54]:

Schoen Gyroid unit cell:(1)$$\emptyset_{G}\left(x,y,z\right)=\sin\left(\frac{2\pi }{a}x\right)\cos\left(\frac{2\pi }{b}y\right)+\sin\left(\frac{2\pi }{b}y\right)\cos\left(\frac{2\pi }{c}z\right)+\sin\left(\frac{2\pi }{c}z\right)\cos\left(\frac{2\pi }{a}x\right)=C$$

Schwarz Diamond unit cell:(2)$$\begin{array}{cc} \emptyset_{D}\left(x,y,z\right)= & \sin\left(\frac{2\pi }{a}x\right)\sin\left(\frac{2\pi }{b}y\right)\sin\left(\frac{2\pi }{c}z\right)\\ & +\sin\left(\frac{2\pi }{a}x\right)\cos\left(\frac{2\pi }{b}y\right)\cos\left(\frac{2\pi }{c}z\right)+\cos\left(\frac{2\pi }{a}x\right)\sin\left(\frac{2\pi }{b}y\right)\cos\left(\frac{2\pi }{c}z\right)\\ & +\cos\left(\frac{2\pi }{a}x\right)\cos\left(\frac{2\pi }{b}y\right)\sin\left(\frac{2\pi }{c}z\right)=C \end{array}$$
where (*x*,*y*,*z*) are the Cartesian coordinate system and *a*, *b*, *c* are the lengths of the unit cells in the *x*, *y*, and *z* directions. In this study, *a*, *b*, and *c* were kept constant to obtain isotropic properties. The constant *C* is the defined relative density. Sheet-based TPMS structures were defined as zero isosurface where the level-set function was ∅ (*x*,*y*,*z*) = 0. To consider having a thickness, the unit cell was enclosed between two isosurfaces, ∅ (*x*,*y*,*z*) = d and ∅ (*x*,*y*,*z*) = −d, where d defines the value of sheet thickness [55].

In considering the resolution of the SLM machines, the sheet thickness of the TPMSs was kept constant at 300 μm, and the unit cell size was varied to generate different pore sizes (600–1500 μm), porosities (54–72%), volumes, and surface areas depending on the structure unit cell type and test specimens. Pore size is defined as the interconnected pore size, which is the diameter of a sphere that is passing through the largest pore of the porous structure (Figure 1a). It has been demonstrated by Maskery et al. [56] and Barnes et al. [57] that a minimum of a 4 × 4 × 4 unit cell repeat is required in every direction to minimize the size effect on the structural mechanical performance of the unit cell. Having a minimum number of unit cells to avoid structural size effect was considered in this study to ensure the potential translation of study findings to larger structures, such as orthopaedic implants. For compression specimens, gyroid scaffolds had a diameter of 11.04 mm and a height of 16.56 mm, whereas diamond scaffolds had a diameter of 14.05 mm and a height of 21.08 mm (Figure 1b). For tensile specimens, gyroid and diamond had a diameter of 11.04 mm and 14.05 mm, respectively, with a parallel length to diameter ratio of at least 2:1 (ISO 6892:2019) [58]. Gyroid and diamond had a gauge length of 14.72 mm and 18.74 mm, respectively, with a gripping length of 20 mm on either side (Figure 1c). For three-point bending specimens, both gyroid and diamond had a width and thickness of 12.5 mm, a span length of 50 mm, and a support span of 43 mm (Figure 1d). Torsion samples were designed according to ASTM E143-13 [59]. The diameters of the round, porous torsion samples were set to be at least five times the unit cell size of the lattice structure with a gauge length-to-diameter ratio of at least 4:1. Both gyroid and diamond samples had a diameter of 14.06 mm, a gauge length of 58.22 mm, and a total length of 86.22 mm. The samples’ top and bottom flat grip sections were each 10 mm long and long enough to be attached to the jaws of the Instron machine (Figure 1e). All the specimens were then manufactured with SLM for morphological and mechanical investigation.

#### 2.1.2. Powder Material

Ti6Al4V-grade 23 ELI powder (supplied by A GE Additive Company and manufactured by Darwin Health Technology Co., Guangzhou, China) was used to fabricate the scaffolds. The morphology of the powder particles was examined by scanning electron microscopy (SEM; Thermo/FEI Quanta 200F, Thermo Fisher Scientific, Waltham, MA, USA). As presented in Figure 2**,** the powder particles had a nearly spherical shape with very smooth surfaces indicating efficient flowability of the particles. The particle size distribution (ASTM B822) was $D_{10}=$ 21 µm, $D_{50}=$ 37 µm, $D_{90}=$ 51 µm with an apparent density (ASTM B417) of 2.38 (g/cm)^3^. The chemical composition of this Ti6Al4V powder was also investigated (ASTM B348), highlighting a very low level of carbon, oxygen, iron, and nitrogen (Table 1).

#### 2.1.3. Manufacturing of Testing Specimens

All testing samples were manufactured from Ti6Al4V alloy (Grade 23 ELI) using an SLM machine (EOS M280, Krailling, Germany). We used the printing parameters optimized by Darwin Health Technology Co. to gain the highest print quality with the most minimal deviation between the designed and printed scaffolds. The details of the laser parameters are outlined in Table 2. The fabricated samples were then removed from the build plate by a wire cutting machine and air was blown to remove any unmelted powder. To enhance the mechanical properties of the lattice structure, the samples were heat-treated at 820 °C in an air environment, with a heating rate of 9 °C/min for 2 h, and then finally cooled to room temperature in a furnace. Sandblasting with quartz sand with a particle size of 50 µm at a pressure of 0.6 MPa was performed to remove the loosely bonded particles.

To ensure reproducibility of the data, five replicates each of both porous compression scaffolds (G600 and D800) were manufactured, compressed, and validated. The physical properties of the experimental samples are outlined in Table 3.

### 2.2. Morphological and Mechanical Characterisations

#### 2.2.1. Morphological Examinations

The fabricated compression samples (G800, G1000, D900, and D1100) were scanned, and their morphologies were characterised using a SkyScan (model 1172, Bruker, MA, USA) high-resolution micro-computed tomography (micro-CT) scanner. The scans were performed with a tube voltage of 102 kV, a tube current of 96 µA, a scan time of 30 min, and a voxel size of 10 × 10 × 10 µm. Each sample was rotated from 0° to 180° in steps of 0.5°, and five images were recorded to obtain an average radiograph image. The micro-CT data were then reconstructed into 2D slices, representing the cross-sectional images of the scaffolds with a commercial software package (NRecon, Skyscan N.V., Kontich, Belgium). The reconstruction process included a beam-hardening correction of 35%, a ring artifact reduction of 10, and lower and upper histogram ranges of 0 and 0.15, respectively. Pore size and strut thickness were measured orthogonally to the build plane in four equally spaced slices of the structure, using the ImageJ software package (National Institutes of Health, Bethesda, Maryland, USA). Within each slice, pore size and strut thickness were measured at 30 different locations. Hence, for each scaffold, there were 120 pore size and 120 strut thickness measurements. Porosity, surface area, and scaffold volume were measured using CTAn software (Skyscan N.V., Kontich, Belgium).

#### 2.2.2. Mechanical Tests

To obtain the mechanical properties of the bulk samples, compressive and tensile test samples were tested in compression (ISO 17340-2014) [60] and tension (ISO 6892:2019) [58], respectively, using an Instron mechanical testing machine (model 5985, 250 kN load cell, Instron, Norwood, MA, USA). For the compressive test, the specimens were placed between two flat, hard, metal machine platens, and only vertical movement was allowed. The bulk compression samples were compressed with a constant displacement of 0.01 mm s^−1^ until failure. The strain was calculated as the displacement of the upper surface in the vertical direction (∆*L*) divided by the initial length of the sample (*L*). Machine compliance was considered and compensated for in all displacement data sets. For the tensile test, the dogbone tensile samples were fixed with a grip length of 20 mm and loaded under tension with a constant displacement of 0.01 mm s^−1^ until failure. Both tests recorded the displacement between the grips at 50 Hz. The compression test was performed on the gyroid and diamond scaffolds according to the standard methods for porous and cellular metallic materials described in ISO 13314:2011 [61] in the same fashion as the bulk compression testing. The compressive modulus ($E_{c}$) was measured as the maximum slope of the elastic region of the compression stress−strain curve. Yield strength ($\sigma_{y}$) was measured by intersecting the stress−strain curve with a 0.2% offset line parallel to the elastic region. The compressive modulus was calculated as follows:(3)$$\mathrm{Compressive}\;\mathrm{modulus}\;\left(E_{c}\right)=\frac{\sigma }{\varepsilon }=\frac{\frac{F}{A}}{\frac{\Delta L}{L}}\;$$
where σ is the stress, *F* is the vertical reaction force, *A* is the initial solid cross-sectional area of the upper surface of the scaffold, ε is the strain, ∆*L* is the displacement of the upper surface in the vertical direction, and *L* is the initial length of the scaffold.

Tensile gyroid and diamond scaffolds were tested according to ISO 6892:2019 [58]. The samples were gripped over a 20 mm length on either side. Tensile testing was conducted using the same machine used for the compression tests. Here, the strain was calculated as the measured displacement divided by the specimen’s initial parallel length ($L_{c}$), while stress (σ) was calculated as the measured load (*F*) divided by the specimen’s solid initial cross-sectional area ($S_{o}$) (Figure 3). Dimensions of the round, porous tensile samples are presented in Table 4. The tensile modulus ($E_{t}$) (maximum slope of the elastic region) and tensile yield strength ($\sigma_{y}$) (intersection of the stress−strain curve with a 0.2% offset line parallel to the linear regression of the initial loading) of the scaffolds were measured from the tensile stress−strain graphs. The tensile modulus was calculated as follows:(4)$$\mathrm{Tensile}\;\mathrm{modulus}\;\left(E_{t}\right)=\frac{\sigma }{\varepsilon }=\frac{\frac{F}{S_{o}}}{\frac{\Delta L}{L_{c}}}\;$$
where σ is the stress, *F* is the vertical reaction force, $S_{o}$ is the initial cross-sectional area of the diameter ($D_{o}$), ε is the strain, ∆*L* is the displacement of the upper surface in the Y direction, and $L_{c}$ is the parallel length (initial length of the scaffold).

Three-point bending was conducted to the ASTM C1674-16 standard [62] using the same material testing machine described in the previous sections. Specimens had a span length ($L_{t}$) of 50 mm and a support span ($L_{s}$) of 43 mm. Bending strain (ε) was calculated based on the measured displacement divided by the specimen’s initial thickness (*d*), and bending stress (σ) was calculated as follows:(5)$$\mathrm{Bending}\;\mathrm{stress}\;\left(\sigma \right)=\frac{3FL_{s}}{2bd^{2}}\;$$
where *F* is the vertical reaction force and *b* is the width of the sample. The bending modulus ($E_{f}$) (maximum slope of the elastic region) and bending strength ($\sigma_{f}$) (intersection of the stress−strain curve with a 0.2% offset line parallel to the linear regression of the initial loading) of the scaffolds were measured from the bending stress−strain graphs.

The torsion test was conducted according to ASTM E143-13 [59]. Samples of each structure type (gyroid and diamond) were placed between two flat jaws, fixed at the bottom, and only torsional movement was allowed from the top jaw. Torsional testing was conducted using an Instron linear torsion mechanical testing machine (model E3000, 3 kN & 25 Nm load cell, Instron, Norwood, MA, USA) under quasi-static conditions at a constant speed of 30 degrees/min (which is high enough to make creep negligible) until failure. Torque (T) (N.mm) and angle of twist (θ) (rad) were measured at 1000 Hz and were used to calculate the shear modulus (G), shear stress (τ), and shear strain (γ), as presented below:(6)$$\mathrm{Shear}\;\mathrm{modulus}\;\left(G\right)=\frac{T\;L}{J\;\theta }\;$$
(7)$$\mathrm{Shear}\;\mathrm{stress}\;\left(\tau \right)=\frac{T\;r}{J}\;$$
(8)$$\mathrm{Shear}\;\mathrm{strain}\;\left(\gamma \right)=\frac{\theta \;r}{L}\;$$
where *J* is the polar moment of inertia of a solid bar (mm^4^):(9)$$J=\frac{\pi }{2}\left(r^{4}\right)\;$$
where *r* is the radius of the bar (mm). Torsional stiffness (maximum slope of the elastic region) and torsional strength (intersection of the stress−strain curve with a 0.2% offset line parallel to the linear regression of the initial loading) were measured from the torsional stress−strain graphs.

### 2.3. Finite Element Modelling

#### 2.3.1. Geometry and Mesh Convergence

Finite element (FE) modelling of all considered structures was done through use of a similar loading regime, as the experimental testing was performed using Abaqus (version 2019, Dassault Systèmes Simulia Corp, Waltham, MA, USA). In brief, the geometry of the designed models was imported into Abaqus in preparation of the finite element analysis (FEA). Mesh convergence was conducted on both gyroid and diamond lattice structures using tetrahedral elements (C3D10), reducing the element size from 2 mm to 0.05 mm, where it was demonstrated that the results were converged within 5% with an element size of 0.065 mm. For the gyroid scaffold tested under compression, tension, three-point bending, and torsion, results converged with about 15.5, 22.5, 76.0, and 83.1 million elements. For the diamond scaffold tested under compression, tension, three-point bending, and torsion, results converged with about 31.6, 46.0, 76.2, and 83.4 million elements.

#### 2.3.2. Material Properties

The stress−strain data obtained from the experimental characterisation of the bulk materials were used as the input parameters for the FE models. Compressive Young’s modulus (E) and yield strength ($\sigma_{y}$) (0.2% offset) were measured to be 35.77 ± 2 GPa and 1012 ± 45 MPa, respectively. Tensile Young’s modulus (E) and yield strength ($\sigma_{y}$) (0.2% offset) were measured to be 95.06 ± 1.5 GPa and 788 ± 11.4 MPa. Plastic stress−strain data were also inputted into the FE models to consider the plastic deformation of the lattice structures. The loading plates in compression testing were assumed to be rigid; as such, their material properties were irrelevant. The Ti6Al4V bulk material was assumed to be solid and homogeneous. The Poisson’s ratio was set as 0.3. Isotropic elasticity and hardening models were used in all simulations.

#### 2.3.3. Loading and Boundary Conditions

For compression testing, two rigid circular plates were created. The nodes at the bottom and top faces of the gyroid and diamond specimens were tied (fixed) to the bottom and top rigid plates in all directions, respectively (i.e., no sliding or separation was allowed). The bottom plate was fixed (Encastre) in all directions and a reference point (RP), which was allowed to only move in a uniaxial direction, was introduced and constrained to the centre of the top plate. This RP allowed us to apply a uniform uniaxial displacement to all the top nodes of the scaffold and eventually deform the lattice structure until it yielded. A vertical displacement of 1 mm on the RP in the negative Y direction was applied with a constant strain rate of 0.1 s^−1^. The respective reaction force (*F*) and displacement (∆*L*) were measured from this single RP node and, as a result, compressive Young’s modulus/stiffness (E) and yield strength ($\sigma_{y}$) (0.2% offset of the linear regression of the initial loading) of the scaffolds were measured from the compression stress−strain graphs.

For tensile testing, the surface of the bottom, round, solid grip section (20 mm) of the model was fixed (Encastre) in all directions, and the surface of the top grip section was coupled to an RP created at the centre of the top surface of the model. The RP was constrained to move only in a uniaxial direction. A vertical displacement of 2.5 mm was applied to the RP in the positive Y direction with a constant strain rate of 0.1 s^−1^. The respective reaction force (*F*) and displacement (∆*L*) were measured from this single RP node and, as a result, tensile Young’s modulus/stiffness (E) and yield strength ($\sigma_{y}$) (0.2% offset of the linear regression of the initial loading) of the scaffolds were measured from the tensile stress−strain graphs.

For the three-point bending test, two semicircle supports and a semicircle loader with a 5 mm radius were created to replicate the experimental model. To allow sliding between the built-in plates of the scaffold and the supports and loader, tangential behaviour with a penalty coefficient of friction of 0.2 and normal behaviour with hard contact were introduced. The loader and supports had a surface-to-surface interaction with the scaffold (finite sliding) with slave “adjustment only to remove overclosure” and allow penetration onto the surface of the built-in plates of the scaffold. An RP was defined on the top surface of the semicircle loader and was coupled with the top surface of the loader. The RP was constrained to only move in a uniaxial direction. A vertical displacement of 1.5 mm was applied to the RP in the negative Y direction with a constant strain rate of 0.1 s^−1^. The respective reaction force (*F*) and displacement (∆*L*) were measured from this single RP node. As a result, the bending modulus ($E_{f}$) and bending strength ($\sigma_{f}$) (0.2% offset of the linear regression of the initial loading) of the scaffolds were measured from the bending stress−strain graphs.

For the torsion test, the surface of the bottom flat grip section (10 mm) of the model was fixed (Encastre) in all directions, and the surface of the top grip section (10 mm) was coupled to an RP created at the centre of the top surface of the model. The RP was constrained to only move about the *y*-axis. A twist of 0.5 rad was applied to the RP, and the respective moment and twist angle were measured from this single RP node. As a result, the torsional stiffness and torsional strength (0.2% offset of the linear regression of the initial loading) of the scaffolds were measured from the torsional stress−strain graphs.

## 3. Results and Discussion

### 3.1. Morphological Deviation of Additive Manufacturing Specimens from Designs

Key morphological characteristics of the manufactured scaffolds, such as pore size, porosity, and strut thickness, were characterised and compared to their designed CAD values. Table 5 summarises the morphological parameters of micro-CT and CAD values for the gyroid and diamond scaffolds. A constant sheet thickness in both the gyroid and diamond scaffolds was used while the pore size and porosity increased with increasing unit cell size. The measured pore size and porosity of the micro-CT data were smaller than the CAD data. However, the measured sheet thickness of micro-CT data was larger than the CAD data for both scaffolds. It is clear from the information presented in Table 5 that the variation in the strut thickness was dependent on the strut angle with respect to the built plane.

Strut thickness and pore size were measured based on both horizontal (θ = 90) and vertical struts (θ = 0), as revealed in Figure 4a. It was demonstrated that as the designed porosity increases, the percentage error in the manufactured porosity decreases. At the same time, the error in the vertical and horizontal struts and pore size are reduced for the gyroid structures but increased for the diamond structures. The standard deviation between the mean values of each of the four slices was measured for pore size and sheet thickness measurements. For gyroid, the standard deviation between the mean values was 3.2 µm and 7.2 µm for vertical and horizontal struts, respectively, and 6.5 µm and 22.2 µm for vertical and horizontal pore size, respectively. For diamond, the standard deviation between the mean values was 4.6 µm and 5.0 µm for vertical and horizontal struts, respectively, and 7.5 µm and 3.3 µm for vertical and horizontal pore size, respectively.

Due to the over melting of the struts, horizontal struts were generally thicker than their designed values and had a greater deviation (about 64% for gyroid and 34% for diamond) when compared to vertical struts. However, since vertical struts can self-support themselves while being printed, their thickness was larger only by a less than 5% (for both gyroid and diamond) in this study and were in good agreement with the designed values. Data in the literature suggests that, as the vertical printing struts approach the horizontal printing struts, the error margin between the designed form and the 3D printed form increases, where the strut thickness is a function of the angle to the 3D printing plane. [34,63]. This is due to the increased number of partially molten powder particles, which are increased on the downward surface of the horizontal struts [64]. To reduce this deviation, some preventive and post-processing actions, such as design compensation strategies prior to printing, chemical etching, electropolishing, and sandblasting, can be taken [34]. Figure 4c,d reveal the percentage error of pore size and thickness measurements for both vertical and horizontal struts. It can be seen that, gyroid scaffolds (G800 and G1000) had a larger thickness percentage error (approximately 64%) for horizontal struts when compared to diamond scaffolds (D900 and D1100), which was approximately 34%.

Many studies have investigated the effect of geometry on osseointegration performance, and it is important to know that the pore size needs to be small enough to allow for initial osteoblast cell colonisation and large enough to initiate vascularisation of the pre-bone tissue [65]. Van Bael et al. [66] suggested that a pore size of 500 μm can improve the initial cell seeding, colonisation, and attachment, whereas a pore size of 1000 μm is more likely to have better vascularisation of host living cells after 2 weeks. Fukuda et al. [67] also recommended a pore size of 500 μm, but Wu et al. [68] suggested a pore size of 700 μm to enhance osseointegration. Overall, for satisfactory bone ingrowth and enhanced osseointegration, the porosity needs to be above 50% with a pore size range of 300–800 μm, which can benefit vascularisation and cell growth simultaneously [34,69]. A more detailed and comprehensive study on the morphological deviation of TPMS gyroid and diamond was undertaken by Naghavi et al. [70] with a wider range of pore sizes and porosities.

### 3.2. Validation of the Finite Element Model

Simulated stress−strain curves for the gyroid and diamond structures under compression, tension, three-point bending, and torsion tests were compared with their respective experimental data (Figure 5). Overall, good agreement in the compression data was found for all samples with a smooth transition from the elastic to plastic region. The general trend that was observed in almost all data was that the FEA results overestimated the Young’s modulus and yield strength values. This can be due to the manufacturing and material defects, such as imperfect geometry, high surface roughness, microporosity, and the offset of the strut axes from their nominal axes. Unsintered powders and surface roughness can act as stress concentrators and initiate cracks, which can potentially reduce the Young’s modulus and yield strength. It is also clear that the compressive and tensile Young’s modulus and yield strength of the scaffolds were lower than those of the bulk Ti6Al4V. The results in Table 6 demonstrate the Young’s modulus, yield strength, and percentage error between the experimental and FEA results for all samples tested under compression, tension, three-point bending, and torsion. Based on these results, the FE model was validated and further used to develop a wider range of porosities for both gyroid and diamond scaffolds. Figure 6 presents the experimental failure images of the mechanical tests.

### 3.3. Finite Element Analysis of the Mechanical Behaviour of Gyroid and Diamond Topologies

#### 3.3.1. Compression

Figure 7a,b illustrate the comparison of the experimental and predicted trend line of the stiffness and yield strength between the gyroid and diamond samples under compression at different pore sizes. Both gyroid and diamond scaffolds demonstrated a decrease in stiffness and yield strength with increasing pore size. For the range of selected pore sizes and porosities, the gyroid structures’ stiffness varied from 4.40–9.54 GPa and the yield strength from 87–179 MPa. The diamond structures had a stiffness from 5.81–9.89 GPa and a yield strength from 106–170 MPa. It was shown that the diamond scaffold was stiffer (by approximately 65%) and stronger (by approximately 48%) than the gyroid scaffold under compression with similar pore sizes. For example, at a pore size of 1000 μm, the gyroid sample had a stiffness of 5.39 GPa and a yield strength of 105 MPa, respectively, while the diamond sample had a stiffness of 8.87 GPa and a yield strength of 155 MPa, respectively. Figure 8a shows an example of stress distribution within the gyroid and diamond scaffolds both with a pore size of 1000 μm under compression. Bobbert et al. [72] performed a similar study comparing TPMS gyroid and diamond scaffolds in compression. They demonstrated that gyroid scaffolds with a porosity range of 52–66% had a stiffness between 4 and 5.8 GPa and a yield strength from 120–225 MPa. Their diamond scaffolds had a porosity range of 44–60% with a stiffness of between 5 and 6.4 GPa and a yield strength from 150–240 MPa. A study by Barba et al. [73] compared both TPMS gyroid and diamond scaffolds with a constant porosity of 75%. They demonstrated that gyroid had a stiffness of 2.3 GPa and a yield strength of 94 MPa, while the corresponding values for diamond were 3.1 GPa and 129 MPa, respectively.

Considering a constant strut thickness of 300 μm and a varying porosity (50–70%), the values of pore size, stiffness, yield strength, and surface area to volume ratio for gyroid and diamond scaffolds are presented in Figure 9. As the porosity increases, the pore size and surface area to volume ratio also increase (Figure 9a,b). At the same porosity (e.g., 60%), the diamond scaffold had a larger pore size (1026 μm) compared to the gyroid scaffold (746 μm). However, the gyroid had a larger surface area to volume ratio (7.24 m^−1^) compared to the diamond (7.02 m^−1^). It is clear that the difference between the pore size and surface area of the diamond and gyroid increases as the porosity increases from 50–70%. Stiffness and yield strengths reduced with increasing porosity, where the diamond scaffold demonstrated to be stronger and stiffer than the gyroid scaffold at the same porosity (Figure 9c,d). It is clear that the difference between the stiffness and yield strength of diamond and gyroid scaffolds decrease as the porosity increases from 50–70%.

When comparing the stiffness of the cortical bone and lattice structures considered in this study (Figure 7a,b), gyroid scaffolds with a pore size smaller than 900 μm (stiffness > 6.14 GPa) and diamond scaffolds with a pore size smaller than 1400 μm (stiffness > 6.27 GPa) were within the acceptable lower range of cortical bone stiffness (6–30 GPa). Comparing the compressive yield strength of the cortical bone, gyroid scaffold with a pore size smaller than 800 μm (yield strength > 133 MPa) and the diamond scaffolds with a pore size smaller than 1200 μm (yield strength > 131 MPa) are within the acceptable range of cortical bone yield strength (125–210 MPa). Hence, for lattice structures to be qualified for both stiffness and yield strength in mechanical compression performance, it can be concluded that the gyroid with a pore size less than 800 μm and the diamond with a pore size less than 1200 μm are needed when designing bone replacement scaffolds. It is worth mentioning that the porosity of the selected pore size of gyroid (600–800 μm) and diamond (900–1200 μm) was between 54–62% and 56.4–64.6%, which is above the minimum 50% requirement for enhanced bone ingrowth in all cases. Since the suggested pore sizes have stiffnesses within the range of those of the cortical bone, stress shielding is less likely to occur, hence, bone resorption is prevented and implant failure may eventually occur [74].

#### 3.3.2. Tension

Figure 7c,d illustrate the comparison of the experimental and predicted trend line of stiffness and yield strength between the gyroid and diamond scaffolds tested under tension with different pore sizes. Both the gyroid and diamond scaffolds revealed a decrease in stiffness and yield strength with increasing pore size. For the range of selected pore sizes, the gyroid scaffolds had a stiffness in the range of 1.78–3.70 GPa and a yield strength in the range of 80–161 MPa. Diamond scaffolds had a stiffness in the range of 1.99–3.32 GPa and a yield strength in the range of 113–177 MPa. It was found that the diamond scaffold was stiffer (by approximately 37%) and stronger (by approximately 74%) than the gyroid scaffold with the same pore size under tension. For example, at a pore size of 1000 μm, the gyroid scaffold had a stiffness of 2.18 GPa and a yield strength of 95 MPa, respectively while the diamond scaffold had a stiffness of 2.99 GPa and a yield strength of 165 MPa, respectively. Figure 8b presents an example of stress distribution within the gyroid and diamond scaffolds, both with a pore size of 1000 μm, under tension. All samples failed within the porous gauge section. Kelly et al. [75] performed tensile tests on TPMS gyroid with a porosity range of 55–85% and demonstrated that the stiffness and yield strength of their specimens were in the range of 2.9–16.9 GPa and 27.3–73.7 MPa, respectively. This is similar to the findings of the present study.

When comparing the tensile stiffness of the cortical bone and lattice structures in this study (Figure 7c,d), the gyroid scaffolds with a pore size less than 1100 μm (stiffness > 2.00 GPa, yield strength > 91 MPa) and the diamond with a pore size less than 1400 μm (stiffness > 2.14 GPa, yield strength > 122 MPa) are qualified and are within and above the acceptable range of cortical bone tensile stiffness (2–16 GPa) and yield strength (77–98 MPa). Unlike stiffness, having greater yield strength than the suggested range of cortical bone would be beneficial in terms of the mechanical performance of the lattice. Hence, for lattice structures to be qualified for both stiffness and yield strength in mechanical tensile performance, it can be concluded that the gyroid with a pore size less than 1100 μm and the diamond with a pore size less than 1400 μm could be considered. The porosity of the selected pore size of gyroid (600–1100 μm) was between 54–69.7% and between 56.4–68.7% for diamond (900–1400 μm), where both are above the required 50% porosity for enhanced bone ingrowth in all cases.

#### 3.3.3. Three-Point Bending

Figure 7e,f illustrate the comparison of the experimental and predicted trend line of stiffness and bending strength between the gyroid and diamond scaffolds tested under bending with different pore sizes. Both the gyroid and diamond scaffolds demonstrated a decrease in stiffness and yield strength with increasing pore size. For the range of selected pore sizes, the gyroid scaffolds had a stiffness in the range of 3.86–8.70 GPa and a bending strength in the range of 202–437 MPa. Diamond scaffolds had a stiffness in the range of 4.62–7.68 GPa and a bending strength in the range of 216–358 MPa. It was found that the diamond scaffold was stiffer (by approximately 39%) and stronger (by approximately 37%) than the gyroid scaffold with the same pore size under bending. For example, at a pore size of 1000 μm, the gyroid scaffold had a stiffness of 5.00 GPa and a bending strength of 242 MPa, respectively. The diamond scaffold had a stiffness of 6.94 GPa and a bending strength of 322 MPa, respectively. Figure 8c presents an example of stress distribution within gyroid and diamond three-point bending scaffolds both with a pore size of 1000 μm.

When comparing the bending stiffness and yield strength of the cortical bone and lattice structures considered in this study, the gyroid scaffolds with a pore size smaller than 1200 μm (stiffness > 3.86 GPa, yield strength > 202 MPa) and the diamond scaffolds with a pore size smaller than 1500 μm (stiffness > 4.62 GPa, yield strength > 216 MPa) are within the acceptable range of cortical bone bending stiffness (3–15 GPa) and yield strength (45–270 MPa). Considering the minimum porosity requirement for enhanced bone ingrowth (50%), the porosity of the selected pore size of gyroid (600–1200 μm) and diamond (900–1500 μm) was between 54–71.7% and 56.4–70.3%, respectively.

#### 3.3.4. Torsion

Figure 7g,h illustrate the comparison of the experimental and predicted trend line of stiffness and yield strength between the gyroid and diamond scaffolds tested under torsion with different pore sizes. Both the gyroid and diamond scaffolds displayed a decrease in stiffness and yield strength with increasing pore size. For the range of selected pore sizes, the gyroid scaffolds had a stiffness in the range of 3.24–6.49 GPa and a yield strength in the range of 113–244 MPa. The diamond scaffolds had a stiffness in the range of 3.24–6.04 GPa and a yield strength in the range of 114–205 MPa. It was established that the diamond scaffold was stiffer (by approximately 37%) and stronger (by approximately 33%) than the gyroid scaffold with the same pore size under torsion. For example, at a pore size of 1000 μm, the gyroid scaffold had a stiffness of 4.00 GPa and a yield strength of 140 MPa, respectively. The diamond scaffold had a stiffness of 5.47 GPa and a yield strength of 186 MPa, respectively. Figure 8d reveals an example of stress distribution within gyroid and diamond torsional scaffolds both with a pore size of 1000 μm.

Cortical bone has a torsional stiffness in the range of 3.1–3.7 GPa and a yield strength in the range of 49–98 MPa, respectively. Figure 7g,h reveal that gyroid scaffolds with a pore size in the range of 1100–1200 μm had a stiffness in the range of 3.24–3.51 GPa and a strength between 113 and 123 MPa. Diamond scaffolds with a pore size in the range of 1400–1500 μm had a stiffness in the range of 3.24–3.60 GPa and a strength in the range of 114–126 MPa. These data highlight that scaffolds that are within the aforementioned pore sizes, based on the considered designs and materials implemented in this study, may have comparable properties to the bone under torsion. The porosity of the selected gyroid (1100–1200 μm) and diamond (1400–1500 μm) scaffolds was between 69.7–71.7% and 68.7–70.3%. The selected porosities of both structures are greater than the required 50%, which suggests enhanced bone ingrowth.

### 3.4. Comparison of Compressive Properties with the Classical Gibson–Ashby Model

The Gibson−Ashby model [76] demonstrates that there is a relationship between the elastic modulus and yield strength of a cellular structure and their relative density. These relationships are known as the Gibson−Ashby model and are represented by the following equations:(10)$$Relative\;Young’s\;Modulus=\frac{E^{*}}{E_{s}}=C_{1}{\left(\frac{\rho^{*}}{\rho_{s}}\right)}^{n}\;$$
(11)$$Relative\;Yield\;Strength=\frac{\sigma^{*}}{\sigma_{s}}=C_{2}{\left(\frac{\rho^{*}}{\rho_{s}}\right)}^{m}\;$$
where $E^{*}$, $\rho^{*}$, and $\sigma^{*}$ are the elastic modulus, density, and yield strength of open cellular structures, respectively, and $E_{s}$, $\rho_{s}$, and $\sigma_{s}$ are the elastic modulus, density, and yield strength of the cellular structure bulk sample, respectively. In this work, $E_{s}$ was measured to be 35.77 GPa and $\sigma_{s}$ was measured to be 1012 MPa for fully dense, additively manufactured Ti6AI4V. The Gibson−Ashby constant values of $C_{1}$, $C_{2}$, n, and m are dependent on the unit cell topology, geometry, and bulk material properties. $C_{1}$ and $C_{2}$ coefficients are predicted by the Gibson−Ashby model to be in the range of 0.1–4 for relative Young’s modulus and 0.1–1 for relative yield strength. In the Gibson−Ashby model, corresponding $C_{1}$ and $C_{2}$ coefficients are 1 and 0.3, and exponent values n and m are defined to be 2 and 1.5, respectively [77,78]. The relative density can be written in terms of the porosity (*P*) [79] of the cellular structure and can be written as follows:(12)$$\frac{\rho^{*}}{\rho_{s}}=1-P\;$$

Figure 10a presents the relative Young’s modulus versus the relative density values. The sheet TPMS gyroid and diamond data points were well fitted to the Gibson−Ashby model straight lines with the coefficients of determination ($R^{2}$) of 0.999 and 0.976, respectively. Figure 10b displays the relative yield strength versus the relative density values. Additionally, in this figure, a well-fitted line for the relative strength−relative density data points of the gyroid and diamond TPMS lattices with the $R^{2}$ of 0.997 and 0.984, respectively, are presented. For both Young’s modulus and yield strength data within Figure 10, all data are within the predicted range of the Gibson−Ashby limit and, as the relative density increases, both relative Young’s modulus and relative yield strength increase. Hence, there is a well-defined correlation between the mechanical properties and the relative density in the elastic regime. Thus, it is now possible to accurately predict the mechanical properties of the sheet TPMS scaffolds by altering their porosities, as long as loading does not exceed the yield strength.

Using the equations in Table 7, the elastic modulus and yield strength of each layer can be predicted, given that each layer of TPMS is considered as a periodic uniform lattice structure.

### 3.5. Failure Mechanism and Deformation Mode

The deformation and failure mechanisms of AM lattice structures under quasi-static compression loading have been investigated [80,81,82,83,84]. Similarly, in the current study, the deformation process of the gyroid and diamond structures under uniaxial compression loading ensue from the following characteristic stages: elastic regime, yielding, post-yielding, plateau, and densification. As can be seen from the compressive stress−strain behaviour of gyroid and diamond displayed in Figure 5a,b, after yielding, the plastic flow stress hardens. This stems from the continuously curved surface of the TPMS structures, which causes a reduction in the plastic stress and strain localisation, and an increase in the structural integrity through the uniform distribution of strain among all surrounding surfaces (see Figure 8a,b). Furthermore, the deformation mechanisms change when the loading conditions change. For instance, from Figure 5c–f, it can be observed that, when the structures are subjected to the tensile, bending, and torsional loading conditions, the stress−strain curve drops sharply at the post-yielding regime (between 7–10% strain for tension, 10–25% strain for bending, and around 6% for torsion). This drop in the flow stress coincides with localised plastic stress and strain at the curved surface of both gyroid and diamond structures (see red areas in Figure 8c,d). Therefore, it can be implied that the deformation and failure mechanisms of materials with hierarchical structures at macro-scales, such as foams, lattice structures, and gyroids, are dependent on the loading direction. Furthermore, these structures have better structural integrity under compressive loading. Structural integrity is also crucial for the implants; the brittle fracture of porous structure in vivo is extremely dangerous, as it may provoke additional damage, such as peri-prosthetic bone fracture. Conversely, if there is pronounced plastic deformation before fracture of the porous structure, it would be more easily detected and remedial action can be taken.

### 3.6. Limitations

The primary limitation of this study was that, due to cost implications, we could not fabricate all the designed samples for the entire range of porosities (54–71.7%). Hence, mechanical testing was not performed on all mentioned samples. However, to mitigate this, the printed samples were validated, and the FE method was performed for all other designed samples to predict their mechanical behaviour. The percentage error in the strut thickness and pore size mismatch between CAD and printed samples was expected to be reduced with larger pore sizes. Another limitation is that we did not perform any fatigue analysis on the designed samples. However, there is a significant amount of research that has studied the fatigue properties of the gyroid and diamond lattice structures. These studies have demonstrated that the endurance limit, which is the stress level for the number of loading cycles before failure, exceeds a certain threshold; for example, 1 × 10^6^ cycles were found to be approximately 20% of the yield stress [72,75]. However, Wang et al. [85] established that the sheet gyroid and diamond can reach endurance limits of up to 80% and 40%, respectively. Another limitation of this study was that we did not perform any permeability tests on the samples. However, many studies have performed permeability tests for the gyroid and diamond lattice structures and have demonstrated that they are in the range of the permeability values reported for trabecular bone [72,86]. They have also revealed that the permeability increases as the porosity of the interconnected pores increases [87]. On the contrary, O’Brien et al. [88] established that, due to frictional forces on the surface of the samples, as the surface area increases (by increasing the porosity), permeability decreases.

## 4. Conclusions

In this study, two different types of Ti6Al4V lattice structures (i.e., TPMS gyroid and diamond sheet with a constant thickness of 300 μm and porosities of 54–71.7% and 56.4–70.3%, respectively) were designed, manufactured, and tested using a range of experimental and numerical methods. The main conclusions and key findings of this study are outlined below:In compression, gyroid with a pore size less than 800 μm (stiffness > 6.96 GPa, yield strength > 133 MPa) and diamond with a pore size less than 1200 μm (stiffness > 7.27 GPa, yield strength > 131 MPa) are within the acceptable lower range of cortical bone stiffness (6–30 GPa) and yield strength (125–210 MPa). At the same pore size, the diamond scaffold is stiffer (by approximately 65%) and stronger (by approximately 48%) than the gyroid in compression.In tension, gyroid with a pore size less than 1000 μm (stiffness > 2.18 GPa, yield strength > 95 MPa) and diamond with a pore size less than 1400 μm (stiffness > 2.14 GPa, yield strength > 122 MPa) are within the acceptable range of cortical bone stiffness (2–16 GPa) and greater than the acceptable cortical bone tensile yield strength (77–98 MPa). At the same pore size, the diamond scaffold is stiffer (by approximately 37%) and stronger (by approximately 74%) than the gyroid in tension.In bending, gyroid with a pore size less than 1200 μm (stiffness > 3.86 GPa, yield strength > 202 MPa) and diamond with a pore size less than 1500 μm (stiffness > 4.62 GPa, yield strength > 216 MPa) are within the acceptable range of cortical bone bending stiffness (3–15 GPa) and yield strength (45–270 GPa). At the same pore size, diamond is stiffer (by approximately 39%) and stronger (by approximately 37%) than gyroid in bending.In torsion, gyroid with a pore size of between 1100 and 1200 μm (3.24 < stiffness < 3.51 GPa, 113 < yield strength < 123 MPa) and diamond with a pore size of between 1400 and 1500 μm (3.24 < stiffness < 3.60 GPa, 114 < yield strength < 126 MPa) are within the acceptable range of cortical bone torsional stiffness (3.1–3.7 GPa) and greater than the acceptable cortical bone torsional yield strength (49–98 MPa). At the same pore size, diamond is stiffer (by approximately 37%) and stronger (by approximately 33%) than gyroid in torsion.Mechanical and morphological deviation between the designed and printed scaffolds are originated from the over melting of the struts, where horizontal and vertical struts are generally thicker than their designed values. Horizontal struts have a greater deviation (about 64% for gyroid and 34% for diamond) when compared to the vertical struts (about 4% for gyroid and 2% for diamond).

Considering all the mechanical and physical properties of the gyroid and diamond TPMS scaffolds and the manufacturing limitation of a minimum strut thickness of 300 µm, we conclude that gyroid scaffolds with a pore size of 500–800 µm and a porosity of 50–62% and diamond scaffolds with a pore size of 700–1200 µm and a porosity of 50–64.6% can be considered as the optimum lattice structure selection for bone replacement applications.

## Figures and Tables

**Figure 1 bioengineering-09-00504-f001:**
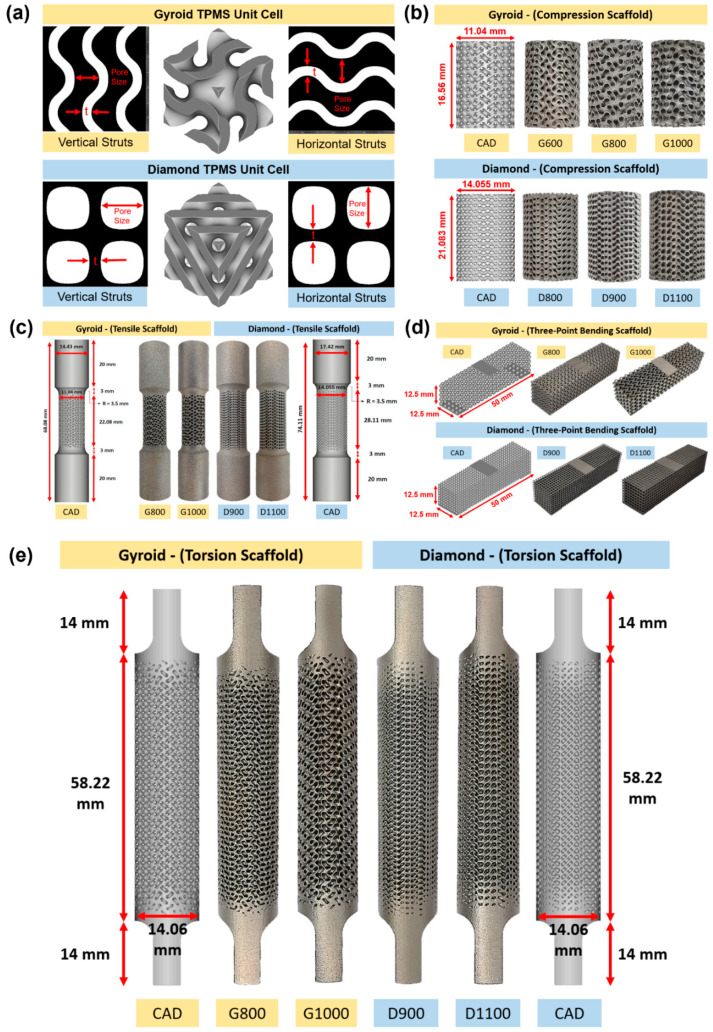
Illustration of (**a**) gyroid and diamond unit cells with defined vertical and horizontal pore sizes and strut thicknesses. Manufactured scaffolds for (**b**) compression, (**c**) tension, (**d**) bending, and (**e**) torsion testing.

**Figure 2 bioengineering-09-00504-f002:**
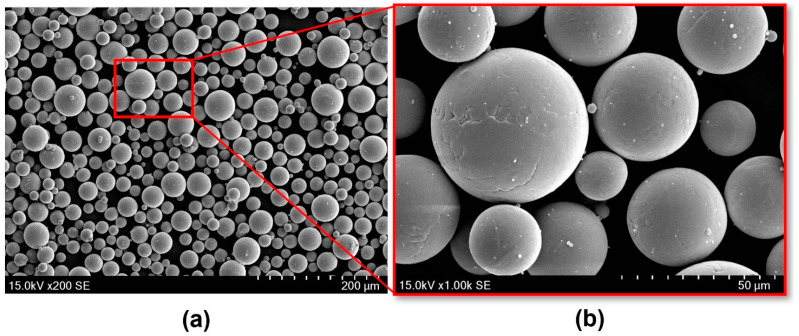
SEM images presenting (**a**) the morphology of spherical Ti6Al4V raw powder for manufacturing lattice structures via SLM and (**b**) powder’s surface appearance.

**Figure 3 bioengineering-09-00504-f003:**
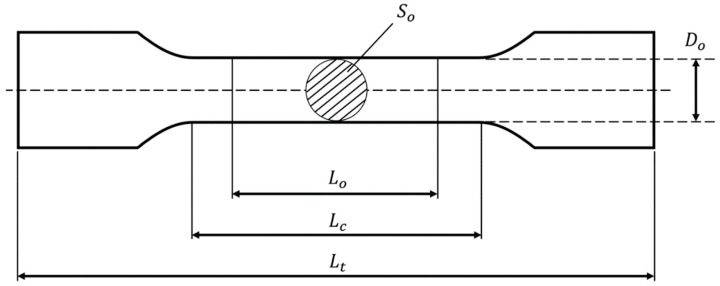
Illustration of the main dimensions on a round, porous dogbone tensile sample.

**Figure 4 bioengineering-09-00504-f004:**
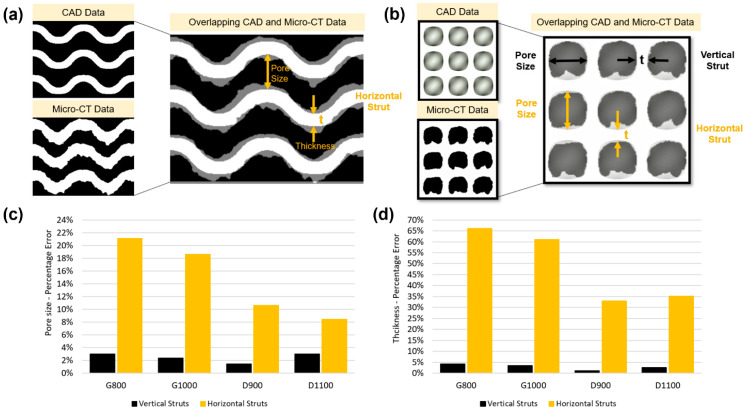
Overlapping CAD and micro-CT morphology data in different strut orientations for (**a**) gyroid and (**b**) diamond scaffolds. Percentage error of (**c**) pore size and (**d**) thickness for both gyroid and diamond scaffolds.

**Figure 5 bioengineering-09-00504-f005:**
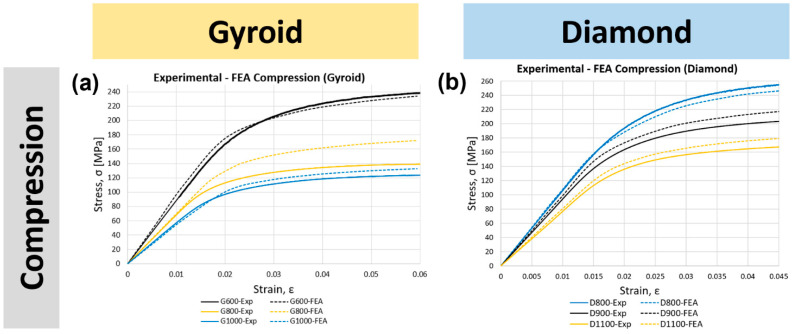
FE model validation of gyroid and diamond scaffolds via experimental mechanical testing in (**a**,**b**) compression, (**c**,**d**) tension, (**e**,**f**) three-point bending, and (**g**,**h**) torsion.

**Figure 6 bioengineering-09-00504-f006:**
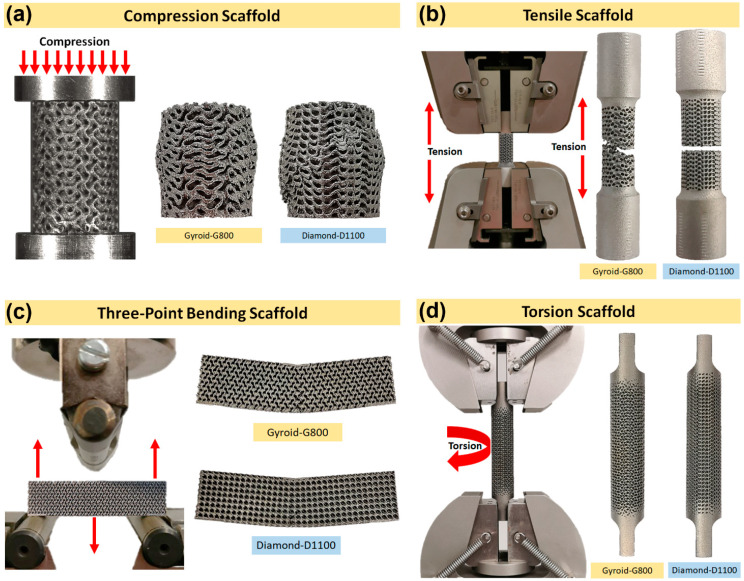
Illustration of the corresponding experimental setup and sample failure mechanism in (**a**) compression, (**b**) tension, and (**c**) bending. (**d**) Torsion samples did not reach the plastic deformation region due to machine load cell limitation and were not broken.

**Figure 7 bioengineering-09-00504-f007:**
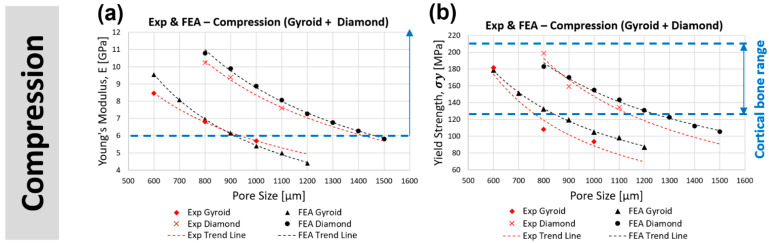
Comparison of FEA-predicted Youngs’s modulus and yield strength of gyroid and diamond scaffolds in (**a**,**b**) compression, (**c**,**d**) tension, (**e**,**f**) three-point bending, and (**g**,**h**) torsion.

**Figure 8 bioengineering-09-00504-f008:**
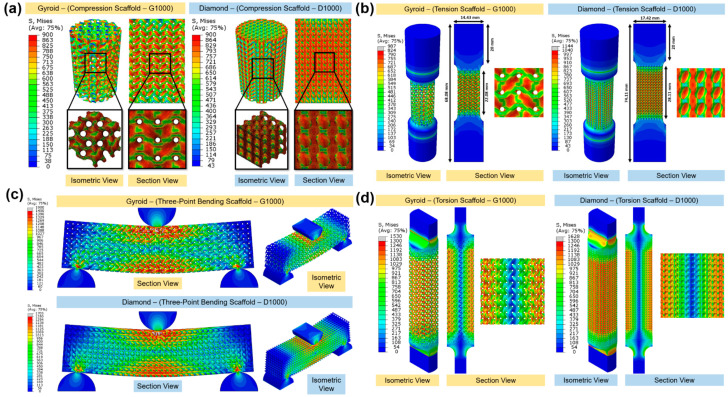
FEA results of gyroid and diamond scaffolds with a pore size of 1000 μm in (**a**) compression, (**b**) tension, (**c**) three-point bending, and (**d**) torsion testing.

**Figure 9 bioengineering-09-00504-f009:**
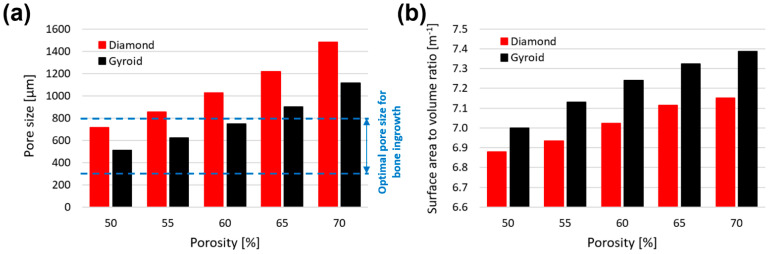
Comparison of the effect of altering porosity from 50–70% on (**a**) pore size, (**b**) surface area to volume ratio, (**c**) Youngs’s modulus, and (**d**) yield strength of gyroid and diamond scaffolds.

**Figure 10 bioengineering-09-00504-f010:**
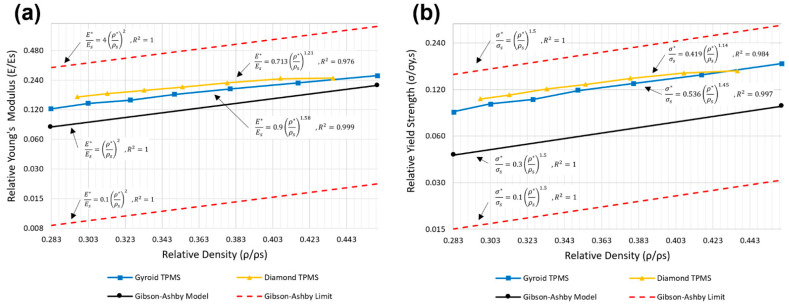
Variation of (**a**) the relative modulus and (**b**) the relative strength of the Ti6Al4V TPMS scaffold with relative density.

**Table 1 bioengineering-09-00504-t001:** Chemical composition of Ti6Al4V powder used in this study.

Element	C	O	N	H	Fe	Al	V	Ti
Standard values (mass %)	≤0.08	≤0.20	≤0.05	≤0.015	≤0.3	≤5.5–6.75	≤3.5–4.5	Balance
Measured values (mass %)	0.01	0.09	0.02	0.0022	0.22	6.44	4	Balance

**Table 2 bioengineering-09-00504-t002:** Laser parameters used in manufacturing Ti6Al4V scaffolds.

Parameter	Laser Power (W)	Layer Thickness (µm)	Scan Speed (mm/s)	Spot Size (µm)	Energy Density (J/mm^3)^	Hatch Distance (µm)
Value	190	30	1000	90	85	110

**Table 3 bioengineering-09-00504-t003:** Physical properties of scaffolds used in the experimental study.

Structure	Scaffold ID	Sheet Thickness (mm)	Pore Size (μm)	Unit Cell (mm)	Porosity (%)
Gyroid	G600	0.3	600	1.808	54.0
	G800		800	2.208	62.0
	G1000		1000	2.608	67.4
Diamond	D800	0.3	800	2.195	52.7
	D900		900	2.400	56.4
	D1100		1100	2.811	62.2

**Table 4 bioengineering-09-00504-t004:** Dimensions of the round tensile samples used in this study.

Scaffold	Total Length (*L_t_*) mm	Parallel Length (*L_c_*) mm	Gauge Length (*L_o_*) mm	Diameter (*D_o_*) mm	Cross-Sectional area (*S_o_*) cm^2^
Gyroid	68.08	22.08	14.72	11.04	95.73
Diamond	74.11	28.11	18.74	14.06	155.26

**Table 5 bioengineering-09-00504-t005:** Morphological parameters for different types of the porous structure. ± indicates the standard deviation of the mean values.

	Pore Size (µm)					Porosity (%)			Sheet Thickness (µm)					Unit Cell Size (mm)
Unit cell	CAD	Micro-CT (Vertical)	% Error	Micro-CT (Horizontal)	% Error	CAD	Micro-CT	% Error	CAD	Micro-CT (Vertical)	% Error	Micro-CT (Horizontal)	% Error	
Gyroid	800	776.2 ± 18.3	3%	631.3 ± 33.8	21%	62.00	54.9	11%	300	312.6 ± 17.3	4%	498.6 ± 37.1	66%	2.208
	1000	976.4 ± 22.4	2%	814.2 ± 35.8	19%	67.40	61.8	8%		310.4 ± 14.5	3%	482.8 ± 24.3	61%	2.608
Diamond	900	887.1 ± 17.3	1%	804.7 ± 16.4	11%	56.40	47.4	13%	300	303.5 ± 13.2	1%	399.1 ± 32.7	33%	2.400
	1100	1066.9 ± 18.9	3%	1007.2 ± 18.9	8%	62.15	55.2	11%		307.5 ± 18.5	3%	405.4 ± 19.5	35%	2.811

**Table 6 bioengineering-09-00504-t006:** Comparison of mechanical properties, obtained via experimental and FE plasticity model study, and human cortical bone properties in compression, tension, bending, and torsion tests [13,14,71,72]. Dash (-) indicates an absence of data.

Test	Sample Name	Young’s Modulus (GPa)			Yield Stress (MPa)		
		Experimental	Simulation	% Error	Experimental	Simulation	% Error
	G600	8.46 ± 0.43	9.54	12%	181 ± 3	179	2%
Compression	G800	6.81	6.96	2%	108	133	23%
	G1000	5.69	5.39	5%	94	105	12%
	D800	10.22 ± 0.31	10.78	5%	199 ± 3	183	8%
	D900	9.37	9.89	6%	159	170	7%
	D1100	7.59	8.06	6%	134	144	7%
	Cortical Bone	6–30	-	-	125–210	-	-
Tension	G800	2.51	2.78	11%	113	122	8%
	G1000	2.39	2.18	9%	99	95	4%
	D900	2.71	3.32	22%	167	177	6%
	D1100	2.67	2.73	2%	132	152	15%
	Cortical Bone	2–16	-	-	77–98	-	-
Three-point bending	G800	6.41	6.32	2%	335	296	12%
	G1000	3.21	5.00	56%	147	242	65%
	D900	7.06	7.68	9%	350	358	2%
	D1100	5.07	6.37	26%	234	287	23%
	Cortical Bone	3–15	-	-	45–270	-	-
Torsion	G800	4.16	4.96	19%	-	171	-
	G1000	3.48	4.00	15%	-	140	-
	D900	4.80	6.04	26%	-	205	-
	D1100	4.23	4.81	14%	-	164	-
	Cortical Bone	3.1–3.7	-	-	49–98	-	-

**Table 7 bioengineering-09-00504-t007:** Comparison of the Gibson−Ashby constant values of $C_{1}$, $C_{2}$, n, and m of gyroid and diamond structures with the classical Gibson−Ashby model.

	Relative Young’s Modulus	Relative Yield Strength
Gibson−Ashby	$\frac{E^{*}}{E_{s}}=1{\left(\frac{\rho^{*}}{\rho_{s}}\right)}^{2}$	$\frac{\sigma^{*}}{\sigma_{s}}=0.3{\left(\frac{\rho^{*}}{\rho_{s}}\right)}^{1.5}$
Gyroid sheet TPMS	$\frac{E^{*}}{E_{s}}=0.9{\left(\frac{\rho^{*}}{\rho_{s}}\right)}^{1.58}$	$\frac{\sigma^{*}}{\sigma_{s}}=0.536{\left(\frac{\rho^{*}}{\rho_{s}}\right)}^{1.45}$
Diamond sheet TPMS	$\frac{E^{*}}{E_{s}}=0.713{\left(\frac{\rho^{*}}{\rho_{s}}\right)}^{1.21}$	$\frac{\sigma^{*}}{\sigma_{s}}=0.419{\left(\frac{\rho^{*}}{\rho_{s}}\right)}^{1.14}$

## Data Availability

All data are contained within the article.

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
