# Peer review of "Mechanical Characterisation and Numerical Modelling of TPMS-Based Gyroid and Diamond Ti6Al4V Scaffolds for Bone Implants: An Integrated Approach for Translational Consideration"

_bioengineering, 2022, doi:10.3390/bioengineering9100504_

Round 1

Reviewer 1 Report

This is a very comprehensive work on the effect of microstructural design on mechanical properties of gyroids and diamond 3D structures to be employed as implants for bone substitutions.

In my opinion this is work will be very useful to the community.

There are only some minor comments:

1)     In 2.1.3 please specify the atmosphere used to post-trat the samples.

2)     Table 5: divide with lines pores size, poristy etc.

3)     Figure 5 and 7, be consistent with the order of EXP and FEA analysis.

4)     Specific surface area is the measurement of surface over weight [m2/g], you here report on m-1. It’s not clear to me the meaning of this. How was it measured?

Author Response

Response to Reviewer 1:

1) In 2.1.3 please specify the atmosphere used to post-treat the samples.

Response: The post-processing was conducted in air environment. This has been added in the context, please reference to line 197 for details.

2) Table 5: divide with lines pores size, porosity etc

Response: Columns have been separated by vertical lines, please reference to Table 5 on page 11 for details.

3) Figure 5 and 7, be consistent with the order of EXP and FEA analysis.
Response: Figure 7 has been updated to be consistent with figure 5. Please reference to Figure 7 on page 18 for details.

4) Specific surface area is the measurement of surface overweight [m2/g], you here report on m-1. It’s not clear to me the meaning of this. How was it measured?

Response: Thanks for pointing this out. Here we are talking about the surface area to volume ratio (m-1) of the scaffold. Figure 9 and text have been updated to ‘surface area to volume ratio’. Please refer to Figure 9 on page 19 for details.

Reviewer 2 Report

Naghavi et al designed and fabricated (using additive manufacturing) two types of porous Ti6Al4V based implants (with gyroid and diamond structures).  Four fabricated samples (G800, G1000, D900, and D1100) were scanned for morphological characterization and mechanical testing included compressive, tensile, bending, and torsional stiffness and strength. Finite element (FE) modelling of all structures was done through use of a similar loading regime. Micro-CT and CAD values were compared for G800, G1000, D900, and D1100 to measure the percentage error of pore size and thickness. Similarly, simulated stress-strain curves under compression, tension, three-point bending, and torsion tests were compared with their respective experimental data. Deviations from simulated data were found for both morphological and mechanical tests.

The study is well conducted and the rationales are well described in the introduction. The description of the methodologies used is thorough. The results have direct relevance the orthopaedic implant design research communities and manufacturers. For better clarity, the authors should:

1.       Remove data repetition. For example, the data on morphological deviation of additive manufacturing specimens from designs are repeated 3 times: in Table 5, in the text (lines 381-389) and on Figure 4. This is confusing and unnecessary, only conclusions and references to literature should be left in the text. Additionally, the same sentences are repeated several times eg ’It is worth mentioning that the porosity of the selected pore size of gyroid (600–800 μm) and diamond (900–1200 μm) was between 54–62% and 56.4–64.6%, which is above the minimum 50% requirement for enhanced bone ingrowth in all cases’ is repeated at least 4 times.

2.       Be more specific which scaffolds were scanned as the scaffolds dimensions were made different for different mechanical tests. Were there any differences in porosity and deviation from designs in relation to the scaffold length (between slices 1 to 4)?

3.       Table 6: were cortical bone values taken from the literature or experimental? In the former, literature references should be added. In the latter, the origin (human? animal?) and the type of the bone tested should be specified.

4.       Discussion: it would be good if the authors were clearer on the intended application of the manufactured scaffolds: is it a critical bone defect or hip replacement stem? Critical-size bone defects occur in both weight and non-weight bearing bones and may have different requirements for mechanical strengths depending on the type of bone to be repaired. Also could the authors summarise their explanations (or hypotheses) on the nature of deviations (both morphological and mechanical) from the designs in the end of the article?

Author Response

Response to Reviewer 2:

  1. Remove data repetition. For example, the data on morphological deviation of additive manufacturing specimens from designs are repeated 3 times: in Table 5, in the text (lines 381-389) and on Figure 4. This is confusing and unnecessary, only conclusions and references to literature should be left in the text. Additionally, the same sentences are repeated several times eg ’It is worth mentioning that the porosity of the selected pore size of gyroid (600–800 μm) and diamond (900–1200 μm) was between 54–62% and 56.4–64.6%, which is above the minimum 50% requirement for enhanced bone ingrowth in all cases’ is repeated at least 4 times.
    Response: Text results were removed from lines 381-389. Figure 4c and 4d represents the percentage error of the pore size and strut thickness. Based on my understanding, this would give a better visual indication of how much larger the error is in horizontal and vertical pore size and struts. Also, since I am showing the deviation between CAD and Micro-CT images of the scaffolds in figure 4a and 4b, it would makes sense to show the percentage error chart below it. We agree with the reviewer that this sentence has been repeated several times throughout the manuscript, although for different parameters. We have now changed the wordings so that it appears less repetitive and are highlighted with yellow color in lines 520-523, 553-556, 578-580 and 603-606.

  2. Be more specific which scaffolds were scanned as the scaffolds dimensions were made different for different mechanical tests. Were there any differences in porosity and deviation from designs in relation to the scaffold length (between slices 1 to 4)?
    Response: Fabricated compression samples (G800, G1000, D900, and D1100) were scanned. This has been addressed in line 209. Standard deviation between the slice mean values has been added and addressed in lines 393 to 399.

  3. Table 6: were cortical bone values taken from the literature or experimental? In the former, literature references should be added. In the latter, the origin (human? animal?) and the type of the bone tested should be specified.
    Response: All the cortical bone values are from the literature and have been referenced in the caption of figure 6 (line 456). All the cortical values of the literature are from human cortical bone. I have also added the ‘human cortical’ bone in the caption.

  4. Discussion: it would be good if the authors were clearer on the intended application of the manufactured scaffolds: is it a critical bone defect or hip replacement stem? Critical-size bone defects occur in both weight and non-weight bearing bones and may have different requirements for mechanical strengths depending on the type of bone to be repaired. Also, could the authors summarize their explanations (or hypotheses) on the nature of deviations (both morphological and mechanical) from the designs in the end of the article?
    Response: We have updated this in lines 123-126 and mentioned that ‘Data obtained in this paper are compared with human cortical bone mechanical properties and it is possible to modulate the design to adapt to particular applications such as large bone defects and load-bearing orthopaedic applications, such as porous hip im-plants that undergo multiple stress states.’ Point 5 was added (lines 753-758) in conclusion to summaise the nature of the deviation of the printed scaffolds from the designs.  

Round 2

Reviewer 1 Report

Authors answered the all points highlighted in the review and the manuscript can now be accepted in the present form.